# Incorporating Neural ODEs into DAE-Constrained Optimization Problems

## Abstract

Differential algebraic equations (DAEs) are pivotal in dynamic optimization across diverse fields, from process control to flight trajectory optimization and epidemiological modeling. Traditional methods like single shooting, multiple shooting, and direct transcription effectively optimize known mechanistic models. However, significant challenges arise when the underlying equations are unknown or deviate from empirical data. While black-box optimization strategies can address some issues, challenges persist regarding data quality, non-linearity, and the inclusion of constraints. Recent advances in machine learning, particularly Neural ODEs, offer promising tools for continuous representation of dynamic systems. This work bridges the gap between machine learning representations of dynamic systems and optimization methodologies, enabling a novel approach for solving DAEs with data-driven components. We demonstrate this approach using numerical examples of DAE problems and realistic case studies, including biochemical reactor control and disease spread prevention. Our results highlight the efficacy of incorporating Neural ODEs into equation-based solvers, showing improved performance over existing strategies such as SINDy. Additionally, we formalize the optimization program for NN-embedded DAEs and present representations for common neural network architectures (e.g., ReLU, tanh). This work contributes a novel framework for dynamic system optimization, integrating machine learning advancements with traditional optimization techniques, and offers practical insights through comprehensive case studies.

## 1 Introduction

### 1.1 Neural Differential Equations

In almost all areas of engineering and fundamental science, the use of differential equation models are ubiquitous. The re-emergence of neural networks (NN) has included extensive research into how these universal approximators can be particularly powerful in their application to differential equation models. Neural ODEs (or NODEs) are a recent breakthrough in the field of computational science and scientific machine learning Chen et al. (2018), where in the most basic sense a neural network learns a non-linear relationship defining the derivatives of system variables given state and parameter values. NODEs are very flexible and build on many of the more traditional numerical methods used for integrating ODEs. Figure 1 shows an example state profile $z$ sampled at various time points $t$.

At any discrete point, a derivative value for $\frac{\mathrm{d}z}{\mathrm{d}t}$ can be computed using a mechanistic equation (e.g. chemical kinetics mass balance), if it is known. Using a integrator scheme such as Euler method or Runge-Kutta, the state at future time points can be computed. In the case of a NODE, the only change is that the mechanistic equation is replaced by a NN function. Figure 1 shows the architecture and basic computation steps.

$$\frac{dz}{dt} = NN(z, u, y, \theta) \tag{1}$$

$$\hat{z} = ODEInt(z_0, t) \tag{2}$$

$$\mathcal{L} = \|z - \hat{z}\|^2 \tag{3}$$

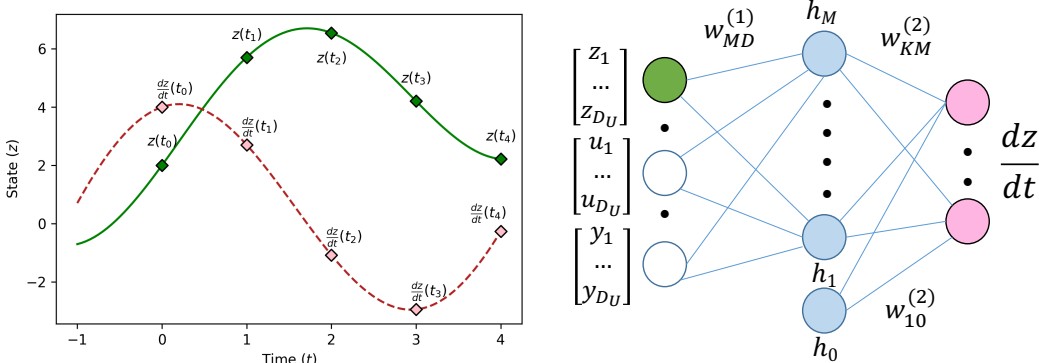

Figure 1: (a) Integration of State Profile with Differential Equation (b) NODE Structure

Recent software packages have allowed a computational graph to be constructed through ODE integrators and thus allow for NN parameters $\theta$ to be learned with state data. Many studies have looked at the application of NODEs for parameter estimation, as well as mechanism discovery Chen et al. (2018) Bradley & Boukouvala (2021). Several research extensions have been presented for NODEs. For a good reference document on NODE applications, methods, and extensions, please refer to Kidger (2022).

### 1.2 OPTIMAL CONTROL OF DYNAMICAL SYSTEMS

A general formulation of a DAE-optimization problem can be presented as follows:

$$
\begin{align}
\min \quad & J(z(t), u(t), y(t)) && (4) \\
\text{s.t.} \quad & \dot{z}(t) = f(z(t), y(t), u(t), t, p), \quad t \in [t_0, t_f] && (5) \\
& g(z(t), y(t), u(t), t, p) = 0, \quad t \in [t_0, t_f] && (6) \\
& z(t_0) = z_0 && (7) \\
& z_L \leq z(t) \leq z_U, \quad y_L \leq y(t) \leq y_U && (8) \\
& u_L \leq u(t) \leq u_U && (9)
\end{align}
$$

In Eq.4, $z$, $y$, and $u$ represent differential, algebraic and state variables respectively, resulting in a DAE model. There has been decades of work focused on solution methodologies particular to these systems of equations. Initial strategies focused on indirect or variational solution methods such as Pontryagin's maximum principle Kopp (1962), which can be inefficient for constrained problems. Later work Sargent & Sullivan (1978) discretized the control profiles and solved via a sequential approach, where the objective and constraint functions are evaluated through forward integration and then gradients with respect to the decision variables are calculated via backwards integration of the adjoint equationsCao et al. (2003). In Bock & Plitt (1984); Bock et al. (2000) and many subsequent works, multiple shooting methods have been explored to address instabilities in single shooting methods. Full discretization of state and control variables creating a large Non Linear Programming (NLP) formulation has been extensively studied as the simultaneous approach Biegler et al. (2002); Nicholson et al. (2018).

### 1.3 MOTIVATION AND CONTRIBUTIONS

While we have many tools for equation-based optimization of known dynamic models, there are still many challenges in applications with complex, unknown, or noisey dynamic phenomena. The work in this manuscript outlines steps to bridging equation-based DAE optimization with ML models such as NODEs. Our contributions can be briefly summarized as:

**1.** We introduce NODEs as surrogates for constrained dynamic optimization problems with unknown or partially known-models, extending their application to areas previously approached with black-box optimization or sparse regression.

**2.** We implement software extensions and demonstrations for incorporating trained NODE models into optimization platform Pyomo Bynum et al. (2021) with common constraint, objective structures and collocation transformations.

**3.** We present case studies with increasing non-linearity, dimensionality, and feasibility complexity to show the strengths and weaknesses in comparison to sparse regression tools.

### 1.4 Related Work

**Sparse Identification of Nonlinear Systems (SINDy).** The goal of SINDy Kaheman et al. (2020) is to identify the sparse set of terms in a function $f$ that best describe the dynamics of the system from input-output data, given a library of candidate functions (solving successive linear or quadratic programs). If an unknown model is recovered, it can be implemented into an equation based optimizer. Other work has extended the sparse regression techniques to nonlinear programming Wilson & Sahinidis (2017) Cozad et al. (2014). Appendix B gives more details of the method.

**Universal Differential Equations (UDEs).** UDEs for ML are introduced in Rackauckas et al. (2020) as part of the SciML software package in Julia. They provide examples of using NN models to extend sparse regression to cases where the underlying model is better represented with a non-linear surrogate. They also provide a suite of software tools for training, optimization and parallelizing code. Some steps of this work for model training/discovery could be done in SciML, but a fully pythonic implementation was maintained for compatibility with existing ML, NLP and DAE tools.

**Black-box optimization.** Several black-box approaches can be leveraged to solve similar dynamic optimization problems through sampling or by building model surrogates Amaran et al. (2016); Larson et al. (2019). Appendix D gives details of popular black-box methods and gives numerical results applying these these methodologies to Case Study 1 (Sec. 3.1), to illustrate these methods.

## 2 Neural Ordinary Differential Equation Augmented Dynamic Optimization: (NODE-ADOpt)

### 2.1 NODE Data and Training

In many industrial applications, sparse, noisy data randomly sampled across time, initial conditions and control actions, may be available. To test our method in the absence of real data mimicking these scenarios, we simulate the system of differential equations with various state and control values. We report number of samples throughout our experiments as the total number of derivative values used to train the NODE in order to control for various approximation schemes for numerical derivatives.

$$p(x) = \frac{1}{\sqrt{2\pi\sigma^2}} e^{-\frac{(x-\mu)^2}{2\sigma^2}}$$ (10)

We also apply noise perturbations to state and control values in order to simulate their effect on the derivatives. We use random noise taken from a normal Gaussian distribution shown in Eq. 10. Here, $\mu$ and $\sigma$ refer to the mean and standard deviation respectively. In Case Study 4, we use real data on the control actions taken during the Covid-19 pandemic and resultant infections.

### 2.2 Algebraic Reformulation and Hybrid Solution

The key step of our approach is to re-integrate the trained NODE into the optimal control problem. Referring to Eq. (4), we can maintain the same overall structure, but the state dynamic constraint is now replaced with the NODE function. Many factors can be considered here when integrating the NN structure with collocation-based optimization methods for optimal control, including NN architecture (size, activation functions,etc), number of time discretizations, and constraint formulations (i.e. complementarity, reduced space, or full-space). To maintain tractability, we aim for surrogates of minimal size that still can achieve an acceptable accuracy. Below, we formalize the resultant mathematical program from integrating NODEs into DAE problems.

Given the general differential equation:

$$\frac{\mathrm{d}z}{\mathrm{d}t} = f(z, u, p) = NN(z, u, W, b), \qquad z(0) = z_0 \tag{11}$$

Example NODE single hidden layer, one state ($z$):

$$\frac{\mathrm{d}z}{\mathrm{d}t} = \sigma(\sum_{i=1}^{n} w_i z + b_i) \qquad \sigma = \text{ReLU, tanh, etc} \tag{12}$$

Applying collocation on finite elements (3 time point example):

$$z_{K+1}(t) = \sum_{k=0}^{K} z_k l_k(t) \tag{13}$$

$$l_k(t) = \prod_{j=0, j \neq k}^{K} \frac{(t - t_j)}{t_k - t_j} \tag{14}$$

$$z_{N+1}(t_k) = z_k \tag{15}$$

For three time points:

$$z_0 \dot{l}_0(t_1) + z_1 \dot{l}_1(t_1) + z_2 \dot{l}_2(t_1) = NN(z_1, W, b) = \sigma(\sum_{i=1}^{n} w_i z_1 + b_i) \tag{16}$$

$$z_0 \dot{l}_0(t_2) + z_1 \dot{l}_1(t_2) + z_2 \dot{l}_2(t_2) = NN(z_2, W, b) = \sigma(\sum_{i=1}^{n} w_i z_2 + b_i) \tag{17}$$

In Eq. 16, we recover 2 equations with two unknown variables ($z_1$, $z_2$). Thus we have an implicit equation for $z(t)$. Several works have developed formal proofs establishing an equivalence to performing fully implicit Runge-Kutta integration of DAE models at Gauss (Radau) points Biegler (2000). Furthermore, collocation on finite elements is a 2K order (2K-1) method which uses K collocation points and can be shown to be algebraically stable Biegler (2010). The constraints defining $NN(z, u, W, b)$ are an important consideration for the ultimate system of non-linear equations, performance with solvers (i.e. convergence, CPU) and also may have effects in the representation of the underlying differential mechanism. For the sake of brevity, we put the mathematical details of explored representations in Appendix A, specifically full-space, reduced-space and complementarity formulations. Complementarity can be used to make ReLU NNs continuous, non-linear constraints and eliminate binaries. Full-space can be applied to non-linear activations and will represent all hidden nodes in the NN as explicit variables, whereas the reduced space builds a mapping directly from inputs to outputs resulting in fewer constraints and variables.

## 3 EXPERIMENTS

### 3.1 CASE STUDY 1: PROOF OF CONCEPT

The first case study is adapted from the Optimal Control code in the Pyomo.DAE repository Nicholson et al. (2018). The problem has 2 coupled ODEs which are functions of control profile $u$. The optimization problem is give in Eq. (18). With known mechanisms, the problem can be solved in PyomoDAE. The corresponding states are shown in Figure 2a using Radau-Lagrange collocation.

$$\min_{u} z_2(t_f) \tag{18a}$$

$$\text{s.t.} \frac{\mathrm{d}z_1}{\mathrm{d}t} = u \tag{18b}$$

$$\frac{\mathrm{d}z_2}{\mathrm{d}t} = z_1^2 + u^2 \tag{18c}$$

$$z_1(0) = 1, z_2(0) = 0, t_f = 1 \tag{18d}$$

To apply the method to this problem, we train a NODE with 3 inputs ($z_1$, $z_2$, $u$) and 2 outputs

$(\frac{dz_1}{dt}, \frac{dz_2}{dt})$. The model is a fully connected sequential NN. Hyperparameters for the hidden layer size, training epochs, and learning rate are tuned. Both ReLU and tanh activations are trained with ADAM optimizer. For the ReLU networks, the NN constraints are formulated using comlementarity. Tanh models are tested with full-space and reduced-space formulations. The problem is solved using 10 finite elements and 2 collocation points for both the know model and for NODE model. Figure 2a shows the model performance using the NODE approach and compares it to the true solution. Profile $z_2$ shows the objective function at $t = t_f$. In Figure 2b, we show the control profile as a function of time with each of the two methods.

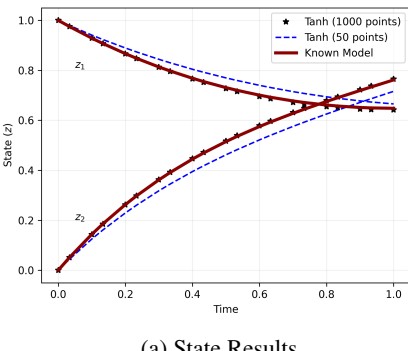

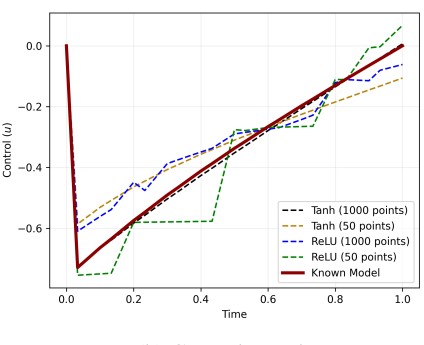

(a) State Results

(b) Control Results

Figure 2: Case Study 1: Comparison of NODE Approach vs Known Model

Figure 2a provides a proof of concept of the NODE-ADOpt method. For low-data scenarios, the fit to the underlying dynamics is slightly worse than in high-data scenarios but still has less than 5% error in its ultimate objective. In Figure 2a, we contrast the control actions from a ReLU NN and tanh NN. While their overall effect on states is similar, the ReLU model results in oscillatory behavior while the tanh model has smooth control actions. Further numerical results are provided for Case Study 1 in Appendix C.1 detailing effects of noise, data-set size, model architecture, and constraint formulation.

### 3.2 CASE STUDY 2: INCLUSION OF PATH-BASED CONSTRAINT

The second case study has 3 coupled ODEs which are functions of control profile $u$. There is a path constraint on $z_2$. Path constraints are important in many dynamic optimization problems but require explicit variables. The formulation of the problem is given in Eq. (19).

$$\min_u z_3(t_f) \tag{19a}$$

$$\text{s.t.} \frac{dz_1}{dt} = z_2 \tag{19b}$$

$$\frac{dz_2}{dt} = -z_2 + u \tag{19c}$$

$$\frac{dz_3}{dt} = z_1^2 + z_2^2 + 0.005u^2 \tag{19d}$$

$$z_2 - 8(t - 0.5)^2 + 0.5 \le 0 \tag{19e}$$

$$z_1(0) = 1, z_2(0) = 0, z_3(0) = 0, t_f = 1 \tag{19f}$$

For this case study, we create a NODE with 4 inputs ($z_1, z_2, z_3, u$) and 3 outputs ($\frac{dz_1}{dt}, \frac{dz_2}{dt}, \frac{dz_3}{dt}$). Similar to the first case study, we explore all activation and constraint representations. After hyperparameter tuning, the number of hidden nodes was set at 50 to balance model size and accuracy. In this case, we use 20 finite elements for both the NODE approach and the base case solution with the known model. This problem is more challenging due to increased dimensionality, and because it requires the satisfaction of a nonlinear constraint as a function of time (Eq. 19e). The flexibility of our NODE

approach allows us to satisfy this nonlinear path constraint. Figure 3a shows the model performance for the state predictions in high and low data scenarios. The state profiles match very closely to the ground truth. We can also observe that the constraint on $z_2$ is satisfied for all $t$. In Figure 3b, we show the corresponding control profiles, that also perfectly match the true optimal control trajectory. Appendix C.2 gives further results for Case Study 2.

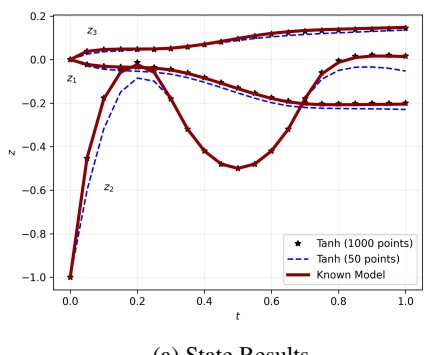
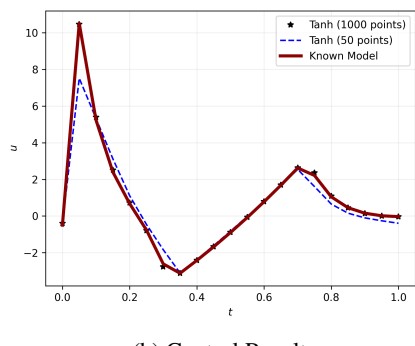

(a) State Results                                      (b) Control Results

Figure 3: Case Study 2: Comparison of NODE Approach vs Known Model

**Solver Results for Different NN Representations** An important research question for using the NODE-ADOpt framework is the effect of different NN constraint representations and how they interact with NLP solvers like IPOPT. In Figure 4, we present summary statistics for the three representations explored.

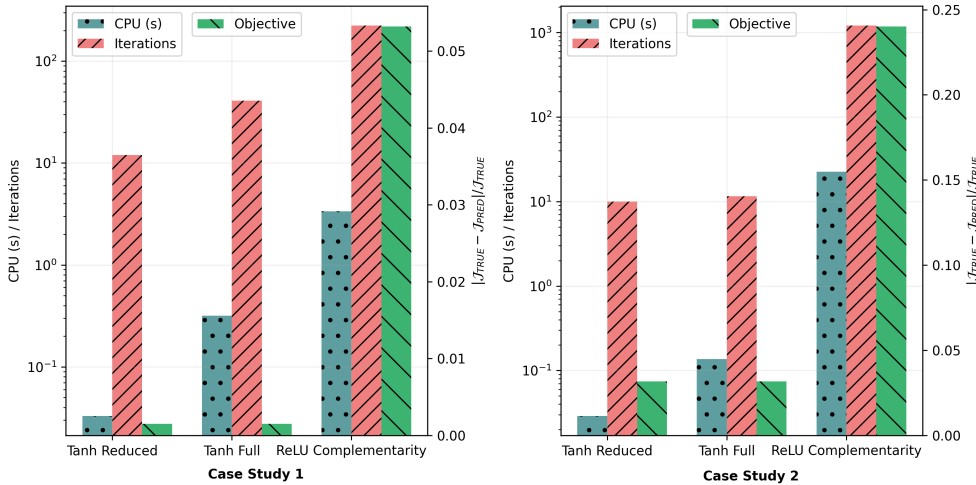

Figure 4: Effects of NN Constraint Representation within DAE

Number of iterations are shown to decrease significantly with using tanh NNs over ReLU for both CS 1 (12 to 223 iters) and CS 2 (11 to 1208 iters). This behavior directly maps onto CPU results with several orders of magnitude reduction throughout cases. Within the tanh models, a reduced-space formulation had superior performance due to fewer constraints and variables in the ultimate optimization model. Outside of speed, the tanh models also found optima that were closer to the known solution. This may be due to ReLU networks' piecewise-linear form which results in some oscillatory behavior around the true control response (shown clearly in Fig. 2b). Due to the consistency in the results, we only use tanh models in subsequent experiments, but applying this framework to novel problems requires some forethought for both the training problem and the dynamic optimization problem.

### 3.3 PARTIALLY UNKNOWN MODEL FED BATCH REACTOR CASE STUDY

The next case study is a more realistic process example of the optimal control of penicillin biosynthesis. The problem has been studied on methods such as evolutionary algorithms, collocation, and dynamic programming Georgakis (2013); Riascos & Pinto (2004). The problem is defined in Eq. (20).

$$\min_{U(t)} \phi = -P(t_f)V(t_f) \tag{20a}$$

$$\text{s.t.} \frac{\mathrm{d}V}{\mathrm{d}t} = \frac{U}{S_F} \tag{20b}$$

$$\frac{\mathrm{d}X}{\mathrm{d}t} = \mu X - \frac{X}{(S_F V)} * U \tag{20c}$$

$$\frac{\mathrm{d}P}{\mathrm{d}t} = \rho X - K_{deg}P - \frac{P}{(S_F V)} * U \tag{20d}$$

$$\frac{\mathrm{d}S}{\mathrm{d}t} = -\mu\frac{X}{Y_{X/S}} - \rho\frac{X}{Y_{P/S}} - \frac{m_S S}{(Km + S)}X + \frac{(1-S)}{S_F}\frac{U}{V} \tag{20e}$$

$$\mu = \mu_{max}\frac{S}{(K_x X + S)}, \quad \rho = \rho_{max}\frac{S}{(K_p + S\frac{(1+S)}{K_{in}})} \tag{20f}$$

$$V(0) = 7, X(0) = 1.5, P(0) = 0, S(0) = 0 \tag{20g}$$

The model is comprised of four differential equations on states: volume ($V$), biomass concentration ($X$), substrate concentration ($S$), and product concentration ($P$). The control variable $U$ denotes the inlet flowrate of substrate. The objective is to maximize penicillin production at final time $t_f$. This study has much more complex dynamics, based on nonlinear kinetic rate laws. This case study is even more challenging, due to its increased dimensionality, nonlinearity and interconnections between all state variables in the model. Moreover, it is important to note that nonlinearity in the parameter space here eliminates the option of employing techniques that assume linearity, such as SINDy. For this problem, we assume we have partial knowledge of the mechanism and we use a NODE to predict change in substrate concentration ($\frac{\mathrm{d}S}{\mathrm{d}t}$). This is common in bio-reactors where inlet reactants and outlet product can be measured but intermediate substrate is difficult to measure directly or formulate a mechanism based on first-principles. In Fig. 5, we solve the optimal control problem, where $U$ becomes a decision variable.

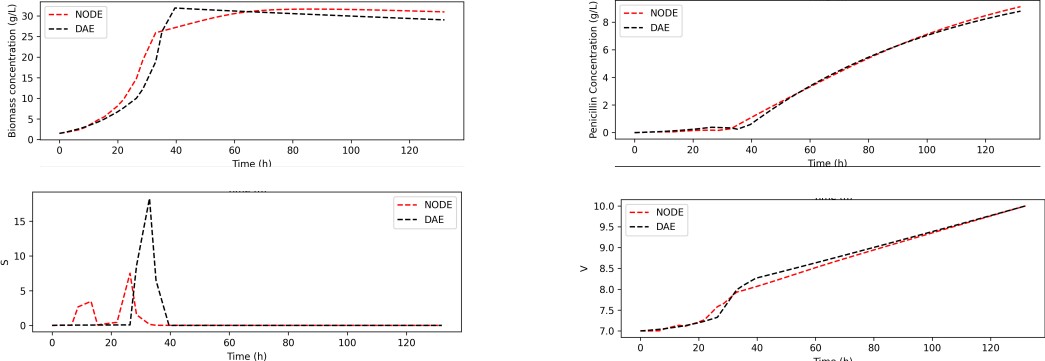

Figure 5: Case Study 3: Comparison of States with NODE Approach vs Known Model

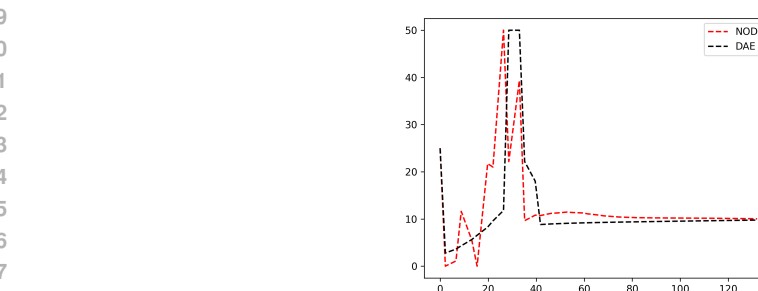

Figure 6: Case Study 3: Comparison of Control with NODE Approach vs Known Model

We solve the NODE-ADOpt problem and reference solution with 20 finite elements and 2 collocation points. While the control profile varies using the NODE approach, the outlet flowrate of penicillin is nearly identical which is the objective function (-91.2 with NODE and -87.9 with known model). Overall, the optimization with the NODE takes 166 iterations in IPOPT for a CPU time of 7.304 seconds.

**Effect of Noise and Comparison with SINDy** Next, we show how NODE-ADOpt compares to popular sparse regression tool SINDy for Case Studies 1-3 with the same data and noise perturbations. We use a polynomial candidate library of order 2 and STLSQ optimizer with a grid search of threshold values to minimize the Bayesian Information Criteria. Further details can be found in B. We import the expressions from the best model into Pyomo to solve the DAE problems.

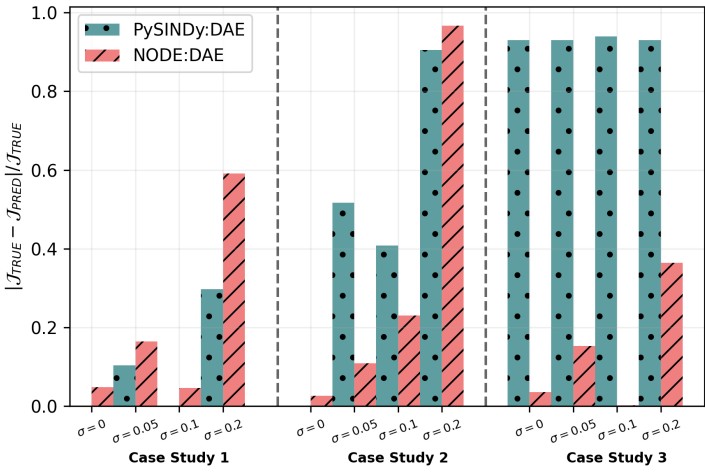

Figure 7: Comparing the Effect of Sample Noise on NODE and PySINDy Recovered Optima

Figure 7 shows the results for how close the predicted optimum ($\mathcal{J}_{PRED}$) is to the true one ($\mathcal{J}_{TRUE}$). For Case Study 1, PySINDy has lower error for all noise levels. This is expected for dynamic mechanisms that are amenable to the SINDy framework: terms that are linear with respect to the candidate function library, parameters with approximately similar orders of magnitude and sparse model representations. For Case Study 2, the results are more mixed. Both approaches seem to give poor results at very high levels of noise ($\sigma = 0.2$) with SINDy doing slightly better. However at lower levels ($\sigma = [0, 0.05, 0.1]$), NODE-ADOpt outperforms the benchmark. In Case Study 3, the limitations of SINDy are shown. Regardless of noise level, the SINDy model selected gives errors of nearly 100%. These results can be explained by the complexity and lack of sparsity in the dynamics. Several terms in Eq. 20 are not recoverable by the method even in ideal circumstances. However, for $\sigma = [0, 0.05, 0.1]$, the NODE gives errors of 3%,15% and 1% respectively.

## 3.4 DISEASE SPREAD PREVENTION CASE STUDY

One final case study looks into optimal quarantine strategies for an infectious disease outbreak. A common model in this field is the QSIR model, which includes 4 differential equations and states: $Q$ population in quarantine, $S$ population of susceptible individuals, $I$ population of infected individuals, and $R$ population of recovered individuals. An exemplar model for this section is presented in Eq. 21, adapted from Nenchev (2020), which is based off the Covid-19 outbreak in Germany before available vaccinations.

$$\min_{U(t)} J = 0.5R(t_f) + Q(t_f) + \alpha \int_0^{t_f} U(t)dt \tag{21a}$$

$$\text{s.t.} \frac{\mathrm{d}S}{\mathrm{d}t} = -b/p * S * I \tag{21b}$$

$$\frac{\mathrm{d}I}{\mathrm{d}t} = b/p * S * I - (m + U) * I \tag{21c}$$

$$\frac{\mathrm{d}R}{\mathrm{d}t} = m * I \tag{21d}$$

$$\frac{\mathrm{d}Q}{\mathrm{d}t} = u * I \tag{21e}$$

$$Q + S + I + R = p \tag{21f}$$

$$Q(0) = 0, S(0) = 80 * 10^6, I(0) = 1000, R(0) = 0 \tag{21g}$$

The states are given as ordinary differential equations. The degree of freedom on the control input $U$ is designed to be the strength of quarantine or lockdown response policy makers put into effect. We adapt the objective function to give a trade-off between minimizing the disease spread (number of individuals recovered $R$) and the cost incurred by the lockdown strength ($U$). To fit the relationship between a general lockdown strength $U$ and real government enforcement mechanisms, we use real infection data from Italy from Riccardi et al. (2021) to fit a NN model. Table 1 gives a summary of the intervention variables provided as inputs to the NN (excluding temperature and humidity). Fig. 8 shows the infection data and NN predictions.

Table 1: Control Variable Descriptions and Bounds

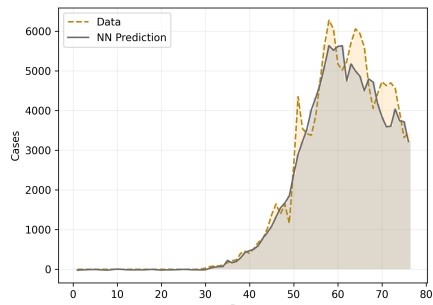

Figure 8: Infection Rate of COVID-19 in Italy

| Variable | Description | Bounds |
|---|---|---|
| $x_1$ | Border Control | [0, 2] |
| $x_2$ | Enforcement | [0, 1] |
| $x_3$ | Testing Volume | [0, 15] |
| $x_4$ | Testing Criteria | [20, 100] |
| $x_5$ | Gov. Mitigation | [1, 10] |
| $x_6$ | Online Presence | [0, 50] |
| $x_7$ | National Flights | [0, 30] |
| $x_8$ | International Flights | [0, 100] |
| $x_9$ | Temperature | [10, 1000] |
| $x_{10}$ | Humidity | [0.1, 1.0] |
| $x_{11}$ | Population Awareness | [0, 24] |

Next, we solve the optimal control problem with the embedded NN predicting $U(t)$ for the QSIR model and objective function. The results are shown in Figure 9.

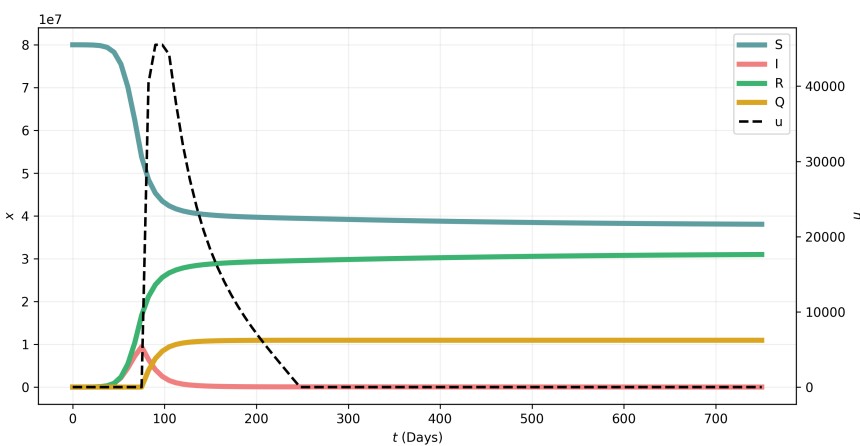

Figure 9: Case Study 4: Solution to Optimal Control Problem

In order to minimize the number of infected, a strong response is required around 80 days. Testing, border enforcement and public awareness ramp up close to their upper bounds. Following this sharp control action, the number of individuals who contract the disease is greatly diminished. While more information is needed to apply this framework to government actions with detailed cost data, this real-world example is another use case for the NODE-ADOpt framework.

## 4 Conclusions and Future Work

This work demonstrates the integration of NODEs into complex, non-linear optimization problems which are constrained by differential and algebraic equations. The methods bridge optimization in an ML paradigm which focuses on model parameter training and traditional constrained non-linear optimization via collocation on finite elements. We show that the resulting large NLPs are algebraically stable and converge quickly for reduced-space smooth representations in numerical experiments. We include examples with increasing non-linearity, problem dimension, and feasibility complexity to show the strengths and weaknesses in comparison to sparse regression and black-box optimization. Some promising directions for future work include investigating non-linear activation functions that have had success in other NODE applications (e.g. SiLU). Furthemore, applying our work to stochastic optimization problems may give insight into how a NODE-DAE structure can handle uncertainty in scenarios or parameter values.

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

# A  MATHEMATICAL FORMULATION OF NEURAL NETWORKS IN OPTIMIZATION PROGRAMS

We denote the input vector by $x \in \mathbb{R}^{N_0}$ and the output vector by $y \in \mathbb{R}^{N_K}$. The input vector to each layer $\hat{z}^{k-1}$ is a linear combination of the output of the previous layer, i.e., $\hat{z}^{k+1} = W^k z^k + b^k$, where $W^k$ is a $N^{k+1} \times N^k$ weight matrix and $b^k$ is a $N^{k+1} \times 1$ bias vector between layers $k$ and $k+1$. Each hidden layer incorporates an activation function $z = \sigma(\hat{z})$, which usually applies a non-linear transformation to each element of the vector input. We denote the vector $x$ as $z^0$ to represent the input layer to the neural network (the input layer is usually not considered a layer). The pre-activation values $\hat{z}^k$ at each layer $k$ are given by (22b) and the post-activation values $z^k$ are denoted by (22c). Finally, the output layer produces the vector $y$ as a linear combination of the final hidden layer given by (22d).

$$z^0 = x \tag{22a}$$

$$\hat{z}^k = W^k z^{k-1} + b^k, \qquad \forall k \in \{1, ..., K-1\} \tag{22b}$$

$$z^k = \sigma(\hat{z}^k), \qquad \forall k \in \{1, ..., K-1\} \tag{22c}$$

$$y = W^K z^K + b^K \tag{22d}$$

It is helpful to express Formulation (22a)-(22d) *element-wise* to demonstrate different neural network representations for the security-constrained optimization problem. Formulation (23a)-(23d) is analogous to (22a)-(22d), but it additionally unfolds the inner layer nodes.

$$x_n = z_n^0 \qquad \forall n \in \{1, ..., N_0\} \tag{23a}$$

$$\hat{z}_n^k = \sum_{i=1}^{N_{k-1}} W_{i,n}^k z_n^{k-1} + b_n^k, \qquad \forall n \in \{1, ..., N_k\}, \forall k \in \{1, ..., K-1\} \tag{23b}$$

$$z_n^k = \sigma(\hat{z}^k), \qquad \forall n \in \{1, ..., N_k\}, \forall k \in \{1, ..., K-1\} \tag{23c}$$

$$y_n = \sum_{i=1}^{N_{K-1}} W_{i,n}^K z_n^{K-1} + b_n^K \qquad \forall n \in \{1, ..., N_K\} \tag{23d}$$

The choice of the best activation function used for Equation (23c) generally falls to the problem of training a neural network, although the ReLU function has been commonly selected for its favorable properties Glorot & Bordes (2011). In the optimization problem, (23a)-(23d) can be implemented using the different aforementioned algebraic representations. As this manuscript utilizes the non-linear ACPF equations, we choose to examine the three following smooth NN representations.

### A.1 FULL SPACE REPRESENTATION (NON-LINEAR)

The variables and constraints described by (23a)-(23d) are formulated explicitly in the problem. The activation constraints can be any smooth function (e.g., tanh, sigmoid, softplus). Intermediate variables (e.g., $z^k, \hat{z}^k$) are formulated in IPOPT and related through sequential constraints.

### A.2 REDUCED SPACE REPRESENTATION (NON-LINEAR)

The reduced-space representation is similar to the full space, but the NN variables and constraints are captured as one expression that connects the input and output variables. Here, intermediate variables (e.g., $z^k, \hat{z}^k$) are not explicitly formulated in IPOPT and the problem variable size is reduced. Previous research has shown that reduced-space representations may have advantages when embedded in optimization formulations Schweidtmann & Mitsos (2019); thus, we implement this formulation to access the advantages over a full-space formulation.

### A.3 ReLU WITH COMPLEMENTARITY REPRESENTATION

While we can formulate ReLU within full- and reduced-space representations, the resulting constraints are not smooth (ReLU is given by $z = max(0, \hat{z})$). To handle this, ReLU can be formulated using complementarity conditions, where we substitute (23c) with (24) for each node in the NN to generate a mathematical program with complementarity constraints (MPCC) Ferris et al. (2002).

$$0 \leq (z_n^k - \hat{z}_n^k) \perp z_n^k \geq 0 \qquad \forall n \in \{1, ..., N_k\}, \forall k \in \{1, ..., K-1\} \quad (24)$$

The complementarity constraints in (24) permit smooth transformations, which have been studied extensively with respect to regularity properties Baumrucker et al. (2008). This manuscript uses a simple component-wise formulation initially presented in Scholtes (2000) given by (25). This representation introduces a non-linear constraint for each node (complementarity) in the neural network and uses the regularization parameter $\epsilon \geq 0$ to satisfy NLP constraint qualifications. This formulation is implemented within `pyomo.mpec` Bynum et al. (2021) and is used in the neural network package `OMLT` Ceccon et al. (2022).

$$(z_n^k - \hat{z}_n^k)z_n^k \leq \epsilon \qquad \forall n \in \{1, ..., N_k\}, \forall k \in \{1, ..., K-1\} \quad (25)$$

Other variations of (25) can be found in the literature and include different regularization techniques, NCP functions, and objective penalties Schewe & Schmidt (2018). Overall, this Section provides a general mathematical framework for embedding NN models from various ML libraries into a non-linear program. Open source code for the tool OMLT can be found online (`github.com/cog-imperial/OMLT`, accessed on 11 July, 2023) for implementing all of the discussed formulations in Python.

## B DETAILS ON MODEL IDENTIFICATION

### B.1 SPARSE REGRESSION PROBLEM DEFINITION

Consider the following dynamical system:

$$\dot{x}(t) = f(x(t), u(t), \theta)$$

where:

- $x(t) \in R^n$ is the state vector,
- $u(t) \in R^m$ is the input vector,
- $\theta \in R^p$ are the model parameters to be identified,
- $f : R^{n+m} \times R^p \to R^n$ is a nonlinear function representing the dynamics of the system.

The goal of SINDy Kaheman et al. (2020) is to identify the sparse set of terms in the function $f$ that best describe the dynamics of the system from input-output data $(u(t), x(t))$, given a library of candidate functions.

The identified model can be represented in the following form:

$$\dot{x}(t) = \sum_{j=1}^{p} \theta_j \phi_j(x(t), u(t))$$

where $\phi_j(x(t), u(t))$ are candidate functions (e.g., polynomials, trigonometric functions, etc.) and $\theta_j$ are the corresponding coefficients.

The identification process involves solving an optimization problem to find the sparse vector of coefficients $\theta$.

## B.2 PROBLEM DATA STRUCTURE

In cases of nonlinear system identification, we typically have a matrix of state observation data shown below as $X$.

$$X = \begin{bmatrix} x_{11} & x_{12} & \cdots & x_{1n} \\ x_{21} & x_{22} & \cdots & x_{2n} \\ \vdots & \vdots & \ddots & \vdots \\ x_{m1} & x_{m2} & \cdots & x_{mn} \end{bmatrix}$$

In systems with a control input, we may also have a vector of manipulated variables $U$.

$$U = \begin{bmatrix} u_{11} \\ u_{21} \\ \vdots \\ u_{m1} \end{bmatrix}$$

The authors of SINDy expanded their tools in Fasel et al. (2021) to incorporate control inputs. In order to perform system identification for the dynamic contexts, we must provide or approximate the derivative data of the system.

$$\dot{X} = \begin{bmatrix} \dot{x}_{11} & \dot{x}_{12} & \cdots & \dot{x}_{1n} \\ \dot{x}_{21} & \dot{x}_{22} & \cdots & \dot{x}_{2n} \\ \vdots & \vdots & \ddots & \vdots \\ \dot{x}_{m1} & \dot{x}_{m2} & \cdots & \dot{x}_{mn} \end{bmatrix}$$

## B.3 APPROXIMATION OF DERIVATIVE VALUES

Given state data $X$, approximation of the derivative values can be approximated using finite difference method. This can present challenges for noisey or sparse data scenarios or situations where the samples are collected in uneven intervals.

$$\frac{df}{dt}(t_i) \approx \frac{f(t_{i+1}) - f(t_i)}{\Delta t}$$

An alternative approach is to fit a model to the state data and then approximate derivative values using collocation points and the surrogate functions derivative value. Some studies have shown benefits of using these collocations methods with B-splines or polynomial functions.

## B.4 MODEL TRAINING

In system identification, there are two objective measures of a good model:

- Model parsimony: models with less complexity are preferred ($|\theta_j|_0$)
- Model accuracy: models with less prediction error are preferred ($||\dot{X} - \dot{X}_t^{model}||_2 / ||\dot{X}_t||_2$)

Typically these terms are added together in the objective function with a scaling parameter enforcing the relative weight of each.

A few different optimizers have been applied, including LASSO Cortiella et al. (2021) and sequential least squares thresholding Zhu & Wang (2019). Other works have done extensive studies into the convergence qualities of the algorithm Zhang & Schaeffer (2019).

For all the case studies shown in this work, we use Sequentially Thresholded Least Squares (STLSQ) algorithm as the optimizer with SINDy. The STLSQ algorithm defaults to performing sequentially thresholded ridge regression, where the objective function ($\|\dot{X} - \sum_{j=1}^{p} \theta_j \phi_j(x(t), u(t))\|_2^2 + \alpha\|\theta_j\|_2^2$) is minimized. The optimization is done by iteratively solving the least squares problem and progressively masking elements of the weight matrix ($\Theta$) that fall below the specified threshold. This process continues until a stable solution is achieved, ensuring that the resulting model is both parsimonious and well-regularized. The hyper parameter 'threshold' of the algorithm was optimized using Scikit-learn's GridSearchCV, by minimizing the Bayesian Information Criterion (BIC) on a validation data set. BIC in terms of the mean squared error (MSE) can be written as follows:

$$\text{BIC} = n \ln(\text{MSE}) + k \ln(n)$$

where ($n$) is the number of data points and ($k$) is the number of non-zero parameters in the identified model.

### B.5 NONLINEAR PROGRAMMING APPROACH

All previously summarized work has relied on least squares based formulation with explicit candidate library functions. This allows for linear or quadratic programming techniques for solving the constrained optimization problem from regularized least squares. An alternative approach is to formulate the problem as a nonlinear program. While this has some drawbacks with respect to convergence guarantees and computational speed, these forms allow more flexibility and nonlinear parameter dependencies. Wilson & Sahinidis (2017) Cozad et al. (2014)

In Lejarza & Baldea (2022) the authors use the nonlinear approach to show how collocation approaches can be directly built into the optimization model (more robust to noise), allow for constraints to be added (from system knowledge), and they show how moving horizon control theory can be applied to these dynamic model identification problem.

## C FULL NUMERICAL RESULTS FOR EXPERIMENTS

### C.1 CASE STUDY 1 RUNS

Here we show additional results for Case Study 1 with varying levels of samples, noise, and different neural network models.

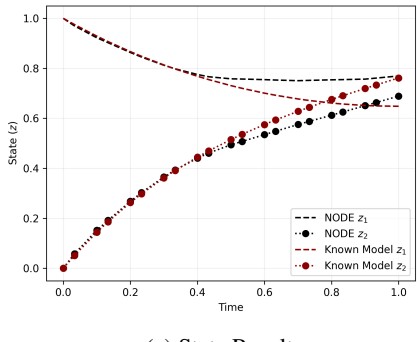

(a) State Results

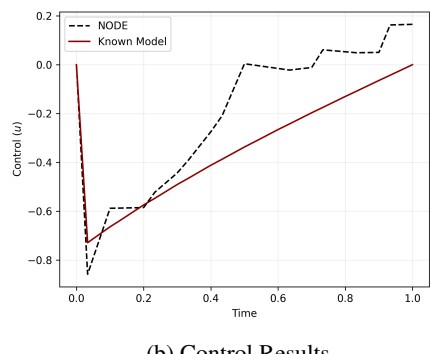

(b) Control Results

Figure 10: Case Study 1: 50 Training Samples with ReLU Complementarity

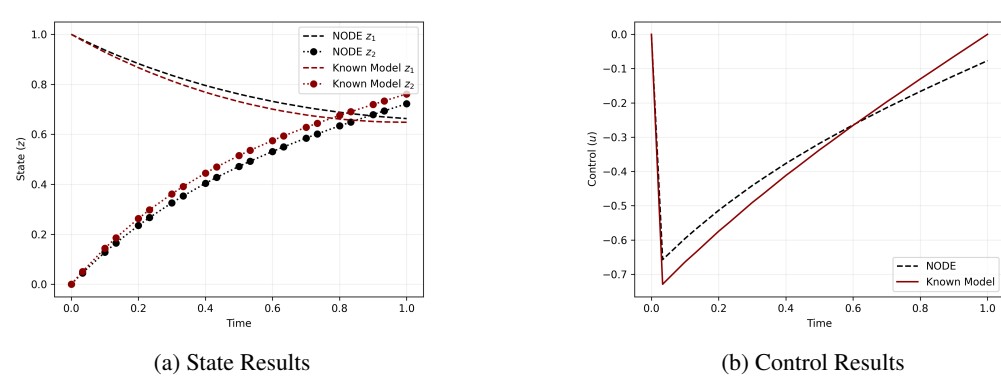

(a) State Results

(b) Control Results

Figure 11: Case Study 1: 50 Training Samples with tanh Full or Reduced space

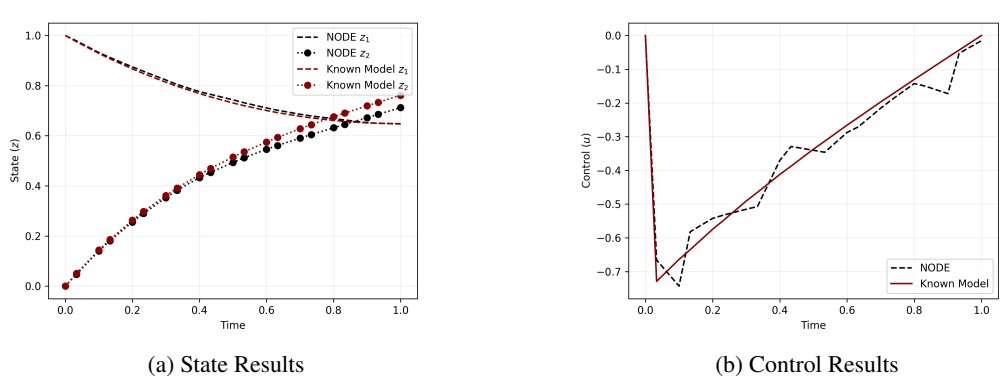

(a) State Results

(b) Control Results

Figure 12: Case Study 1: 100 Training Samples with ReLU

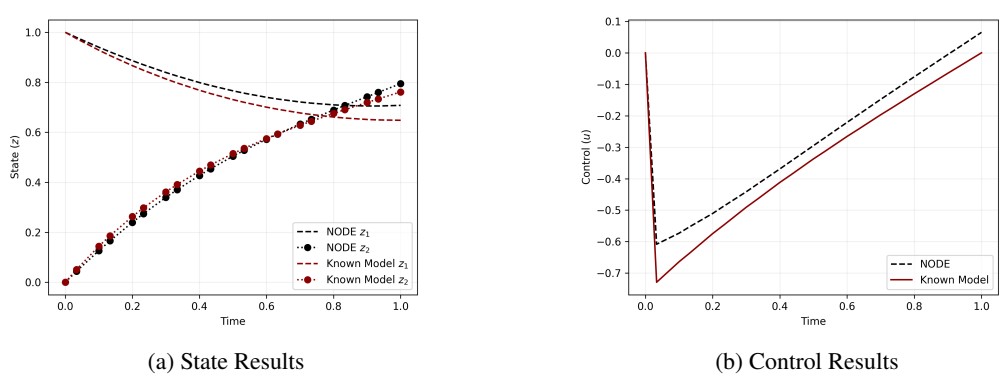

(a) State Results

(b) Control Results

Figure 13: Case Study 1: 100 Training Samples with tanh

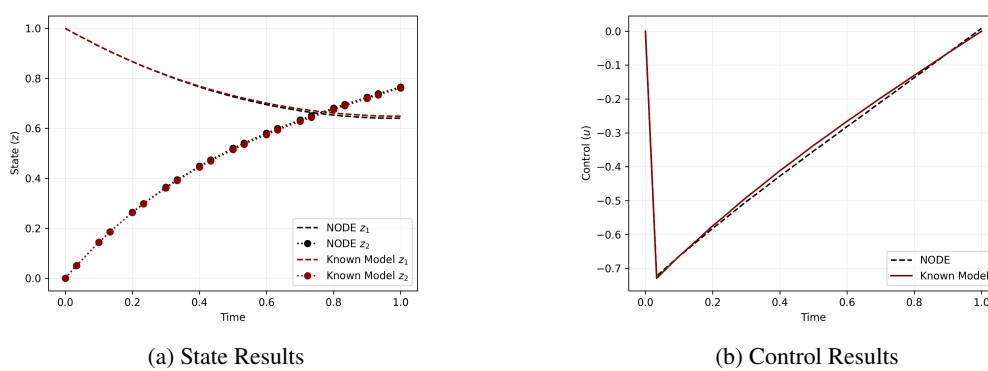

(a) State Results

(b) Control Results

Figure 14: Case Study 1: 1000 Training Samples with tanh

Table 2: Case Study 1: Training and Validation MSE for Tanh and ReLU with Different Training Samples

| Number of Samples | tanh | | ReLU | |
|---|---|---|---|---|
| | $MSE_{Train}$ | $MSE_{Val}$ | $MSE_{Train}$ | $MSE_{Val}$ |
| 50 | 8.75E-04 | 0.0078 | 2.78E-04 | 0.0158 |
| 100 | 5.85E-04 | 0.0018 | 3.92E-04 | 0.0044 |
| 1000 | 4.62E-05 | 5.01E-05 | 7.68E-05 | 9.09E-05 |

Table 3: Case Study 1: Performance Metrics for Tanh and ReLU Models

| Number of Samples | Tanh Full-Space | | Tanh Reduced-Space | | ReLU Complementarity | |
|---|---|---|---|---|---|---|
| | CPU/iters | Obj | CPU/iters | Obj | CPU/iters | Obj |
| 50 | 0.287 / 41 | 7.22E-01 | 0.016 / 13 | 7.22E-01 | 3.12 / 220 | 6.88E-01 |
| 100 | 0.185 / 29 | 7.94E-01 | 0.081 / 10 | 7.94E-01 | 2.886 / 218 | 7.12E-01 |
| 1000 | 4.79E-01 / 53 | 7.65E-01 | 0.002 / 13 | 7.65E-01 | 4.04 / 232 | 7.63E-01 |
| Known Model | 0.009 / 11 | 7.62E-01 | | | | |

## C.2 CASE STUDY 2 RUNS

Here we show additional results for Case Study 2 with varying levels of samples, noise, and different neural network models.

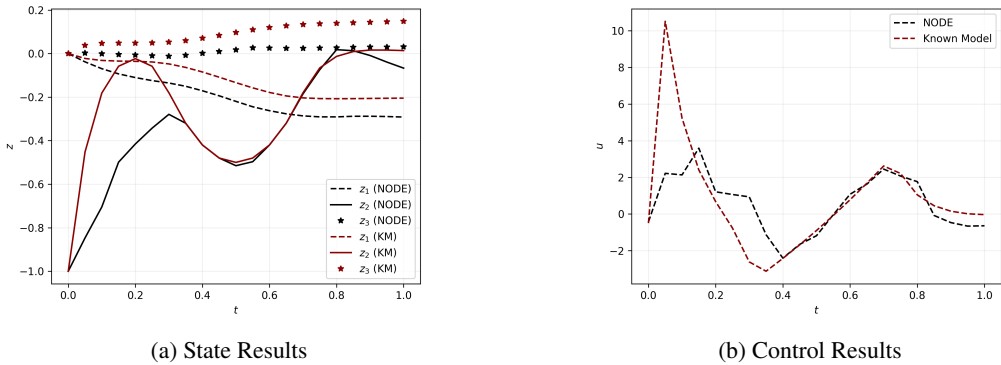

(a) State Results

(b) Control Results

Figure 15: Case Study 2: 50 Training Samples with ReLU Complementarity

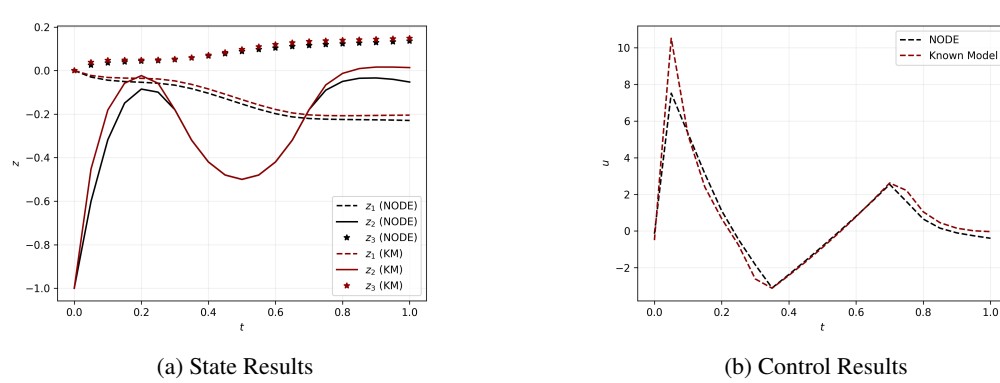

(a) State Results         (b) Control Results

Figure 16: Case Study 2: 50 Training Samples with tanh Full or Reduced space

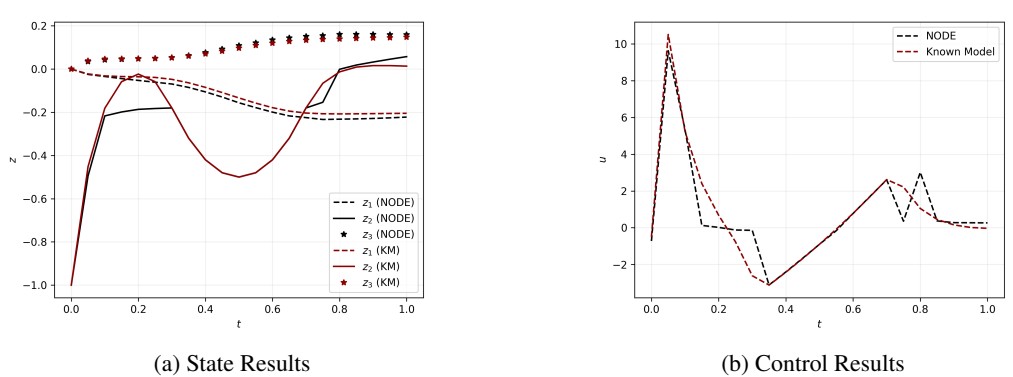

(a) State Results         (b) Control Results

Figure 17: Case Study 2: 100 Training Samples with ReLU

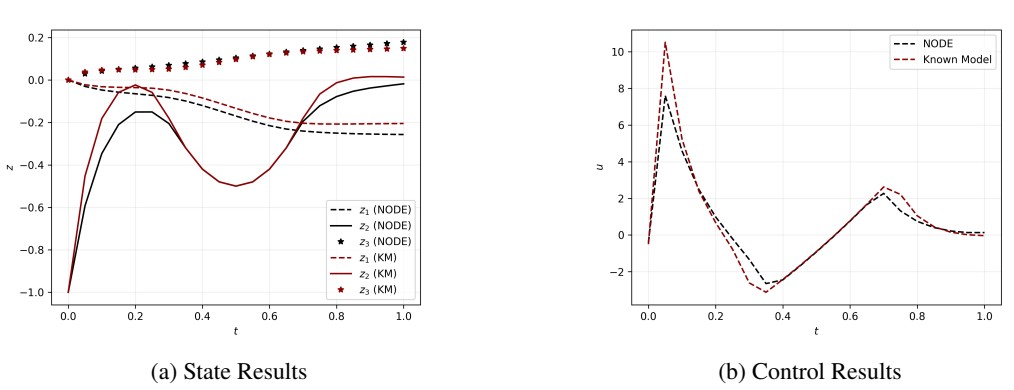

(a) State Results         (b) Control Results

Figure 18: Case Study 2: 100 Training Samples with tanh

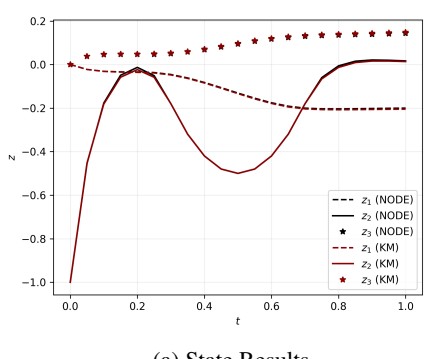 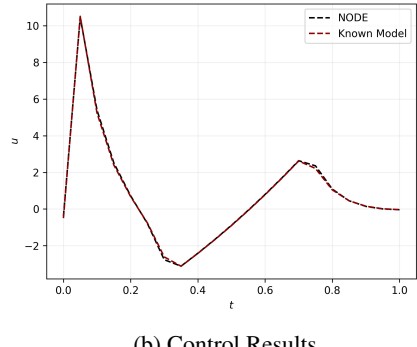

(a) State Results          (b) Control Results

Figure 19: Case Study 2: 1000 Training Samples with tanh

Table 4: Case Study 2: Training and Validation MSE for Tanh and ReLU with Different Training Samples

| Number of Samples | tanh | | ReLU | |
|---|---|---|---|---|
| | $MSE_{Train}$ | $MSE_{Val}$ | $MSE_{Train}$ | $MSE_{Val}$ |
| 50 | 4.52E-04 | 0.0057 | 1.6E-03 | 0.005 |
| 100 | 3.9E-03 | 0.005 | 7.11E-04 | 0.0017 |
| 5000 | 5.24E-05 | 6.98E-05 | 1.27E-04 | 1.14E-04 |

Table 5: Case Study 2: Performance Metrics for Tanh and ReLU Models

| Number of Samples | Tanh Full-Space | | Tanh Reduced-Space | | ReLU Complementarity | |
|---|---|---|---|---|---|---|
| | CPU/iters | Obj | CPU/iters | Obj | CPU/iters | Obj |
| 50 | 0.155 / 13 | 1.35E-01 | 0.03 / 10 | 1.35E-01 | 21.36 / 971 | 3.11E-02 |
| 100 | 0.133 / 11 | 1.77E-01 | Did Not | Converge | 24.71 / 1554 | 1.59E-01 |
| 1000 | 0.119 / 11 | 1.43E-01 | 0.028 / 10 | 1.43E-01 | 21.37 / 1099 | 1.45E-01 |
| Known Model | 0.009 / 11 | 1.47E-01 | | | | |

## C.3 CASE STUDY 3 RUNS

In this section, we train separate NODEs for each state in order to scale each effectively based on validation error. We do this to show NODEs ability to validate the full penicillin model, while in Sec. 3.3 we assume partial knowledge of the mechanism. After testing several sized models, we find that 10 hidden nodes is sufficient to train $V$. For $X$, $P$ and $S$ we must use two hidden layers with 20 nodes each. After training, we first validate the model performance using a known step function input for control profile $U$ and integrate forward using the NNs and the known DAE model. We integrate each using 4th order Runge-Kutta method. The results are shown below.

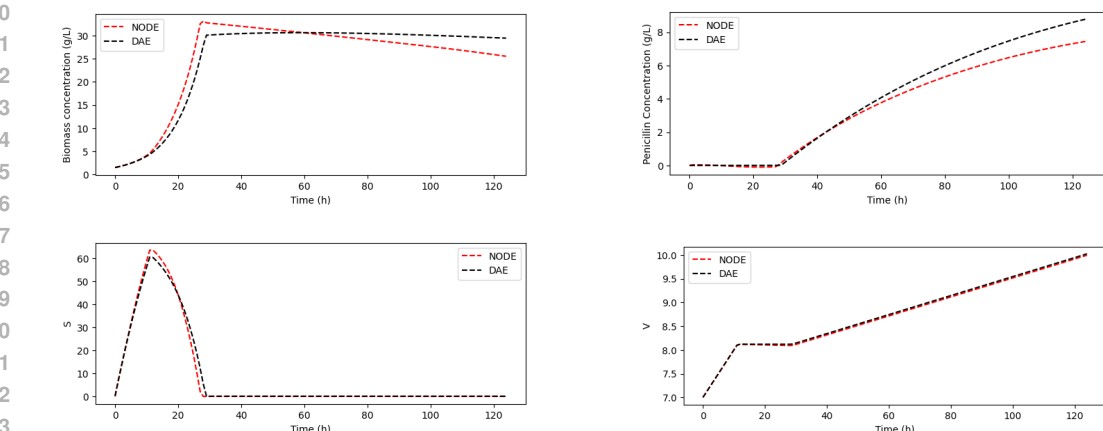

Figure 20: Fed Batch Reactor Validation

While each state is dependent on each other for its dynamic profile, it is clear that $V$ matches almost perfectly due to the simple linear relationship. $S$ fits very well even though the it has the most complex function it must learn. $X$ and $P$ fit well except for some slight deviation toward the end of the time profile.

Table 6: Case Study 3: Training and Validation MSE for Tanh with Different Training Samples

| Number of Samples | Tanh | |
|---|---|---|
| | $MSE_{Train}$ | $MSE_{Val}$ |
| 80k-dP | 5.802E-06 | 2.59E-06 |
| 80k-dS | 9.76E-04 | 6.04E-04 |

Table 7: Case Study 3: Performance Metrics for Tanh Full-Space Models

| Number of Samples | CPU/iters | Obj |
|---|---|---|
| 80k-dP | 1.947/367 | 1.048E02 |
| 80k-dS | 7.304/166 | 9.12E01 |
| Known Model | 1.086/461 | 8.79E01 |

## C.4 RESULTS WITH NOISE PERTURBATIONS

In this section we show results for each case study when we add random noise.

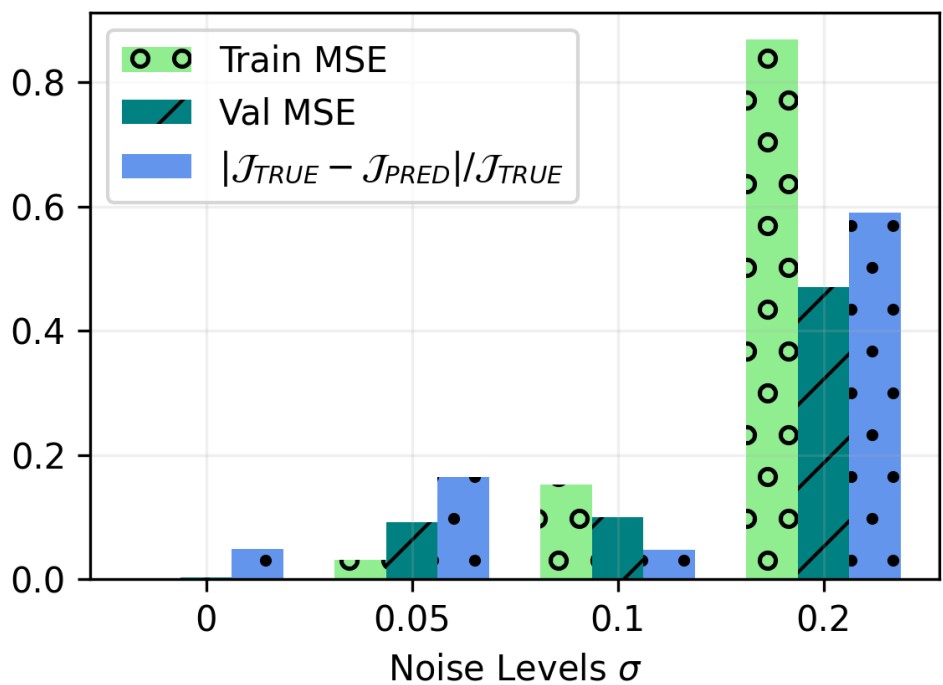

Figure 21: Case Study 1: Noise Perturbations

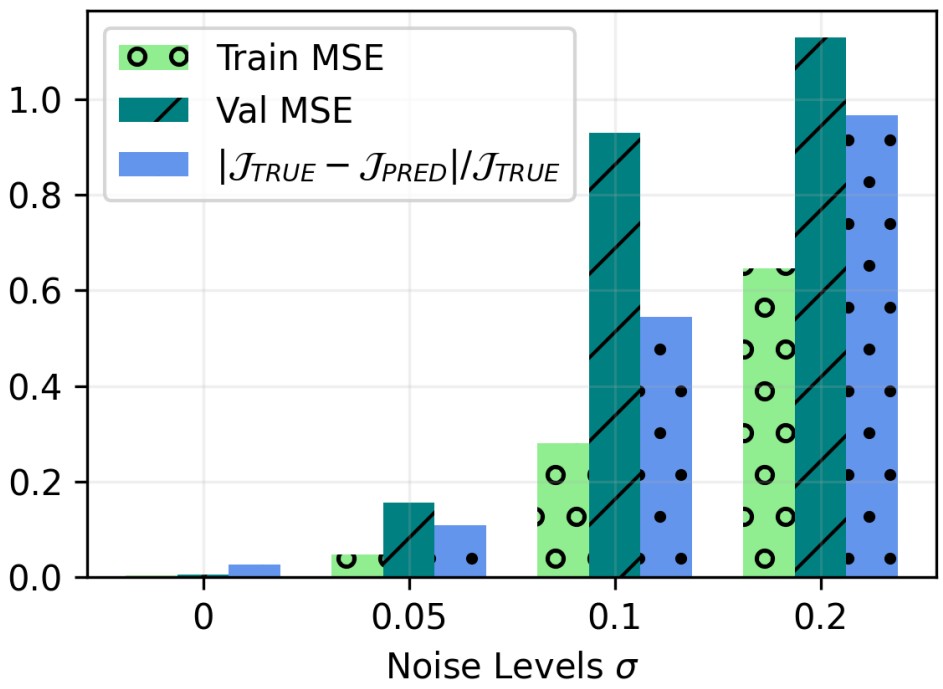

Figure 22: Case Study 2: Noise Perturbations

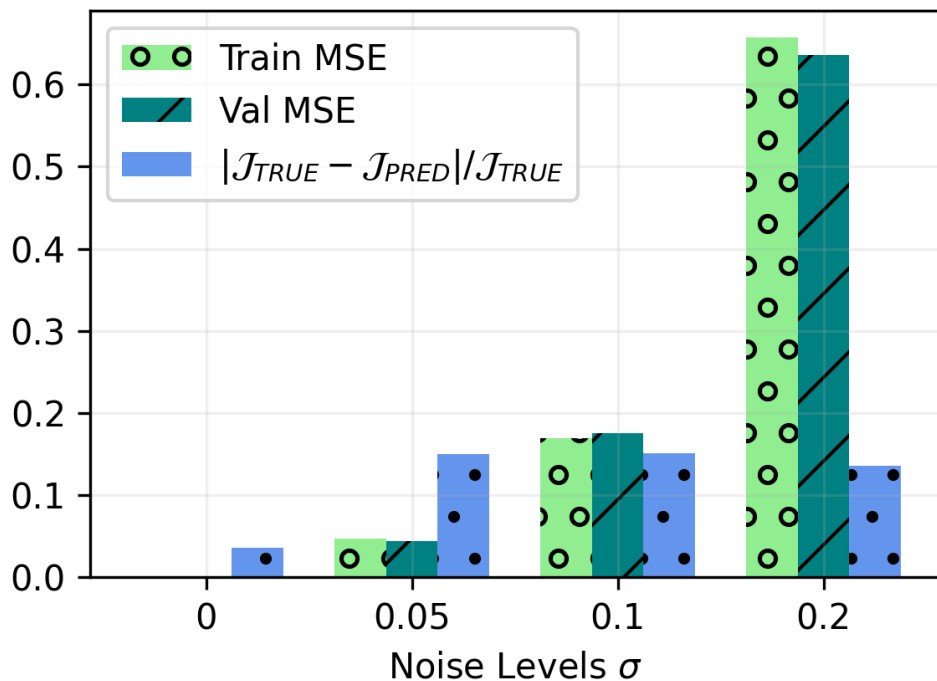

Figure 23: Case Study 3: Noise Perturbations

## D    DETAILS ON BLACK-BOX OPTIMIZATION

Black-box optimization techniques have gained prominence in dynamic systems and control, particularly when analytical solutions are impractical. Recent advances leverage surrogate models and machine learning to improve optimization efficiency. For instance, Lye et al. Lye et al. (2021) employed deep neural networks for PDE-constrained optimization, introducing new training procedures for time-dependent parametric PDEs. Beykal et al. Beykal et al. (2020) developed a data-driven optimization algorithm for differential-algebraic equations (DAEs), utilizing surrogate models to navigate numerical feasibility boundaries. Building on this approach, they later presented a data-driven optimization framework for time-varying systems Beykal et al. (2022), which derives optimal continuous-time control trajectories using surrogate modeling without fully discretizing high-fidelity models. Similarly, Qiu et al. Qiu et al. (2020) proposed a deep learning-based surrogate to replace computationally intensive DAE constraints in operational planning.

Other notable contributions include Moya and Lin's Moya & Lin (2023) DAE-PINN framework, which uses physics-informed neural networks for simulating semi-explicit DAEs. Shokry and Espuña Shokry & Espuña (2014) introduced a sequential dynamic optimization method using kriging metamodels for highly nonlinear processes. Deshmukh and Allison Deshmukh & Allison (2017) presented Derivative Function Surrogate Modeling (DFSM) to approximate state derivatives with surrogate models, streamlining dynamic system design. These approaches, alongside other surrogate-assisted optimization techniques Beykal et al. (2022); Djeumou et al. (2022); Del Rio-Chanona et al. (2019); Safdarnejad et al. (2019); Rall et al. (2020); Kudela & Matousek (2022), highlight the effectiveness of integrating data-driven methods with physical insights in optimizing dynamic systems and control.

Alternatively, when the dynamic system is available as a simulator, the optimal control problem can be directly tackled using data-driven optimization techniques. This approach requires discretizing the control profile over time and treating these discretized values as decision variables for the optimizer. However, this can lead to a high-dimensional problem depending on the number of discretization points used in the control profile. To compare this method with surrogate-based approaches, we apply

this data-driven optimization strategy to Case Study 1. For consistency, we set the number of finite elements to 10, resulting in 11 discretization points for the control profile u(t) with u(0) = 0. We evaluate seven different data-driven solvers for this task: Bayesian optimization, genetic algorithm, particle swarm optimization, and simulated annealing using the package Blanke (2024).

Since the discretization points are treated as decision variables by the solver, a purely data-driven optimization approach can lead to variability in the solution due to the complexity and the non-linearity of the search space. To evaluate the performance and consistency of the different solvers, we conducted ten independent optimization runs for each algorithm. For all solvers, the maximum number of iterations or samples was set to 3000, utilizing the default settings. Additionally, an early stopping criterion based on a patience set of 300 iterations was applied to facilitate termination when appropriate. For each run, we recorded the optimized objective function value $J_{PRED}$ and compared it to the true optimal value $J_{TRUE}$ obtained from the equation based model. The relative error for each run was calculated as a normalized measure of the deviation of the predicted optimal value from the true value, allowing for a fair comparison across different algorithms. We aggregate the relative errors for all ten runs of each optimizer to compute the mean and standard deviation, which reflect the accuracy and robustness of each method, respectively. The results are summarized in Figure 24, which showcases the average relative errors for each optimization algorithm, with error bars representing one standard deviation. This figure highlights both the accuracy and consistency of the methods evaluated.

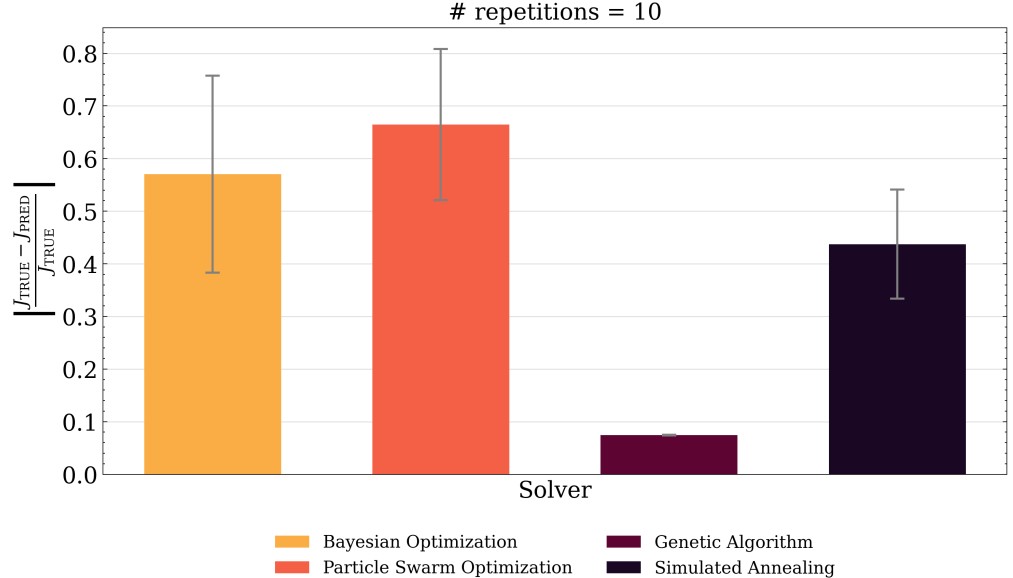

Figure 24: Average relative errors of optimization algorithms with standard deviation error bar

Figure 25 presents the computational time required by each optimizer, providing insights into their efficiency. Figure 26 illustrates the evolution of the optimal solution found by each solver as a function of the number of samples collected or utilized, demonstrating how quickly each method converges towards the optimum.

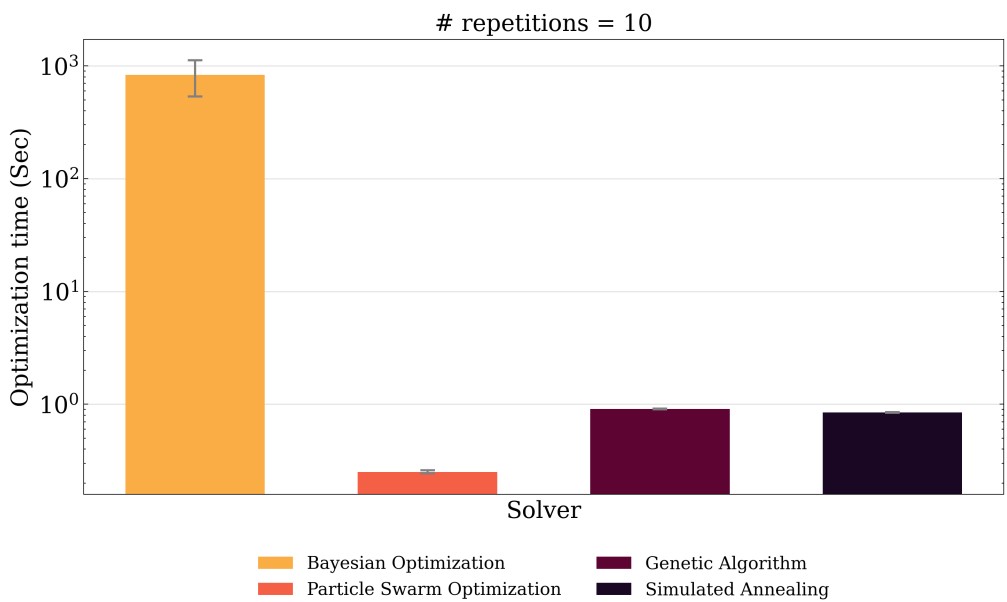

Figure 25: Computational time requirements (sec) of each optimization algorithm

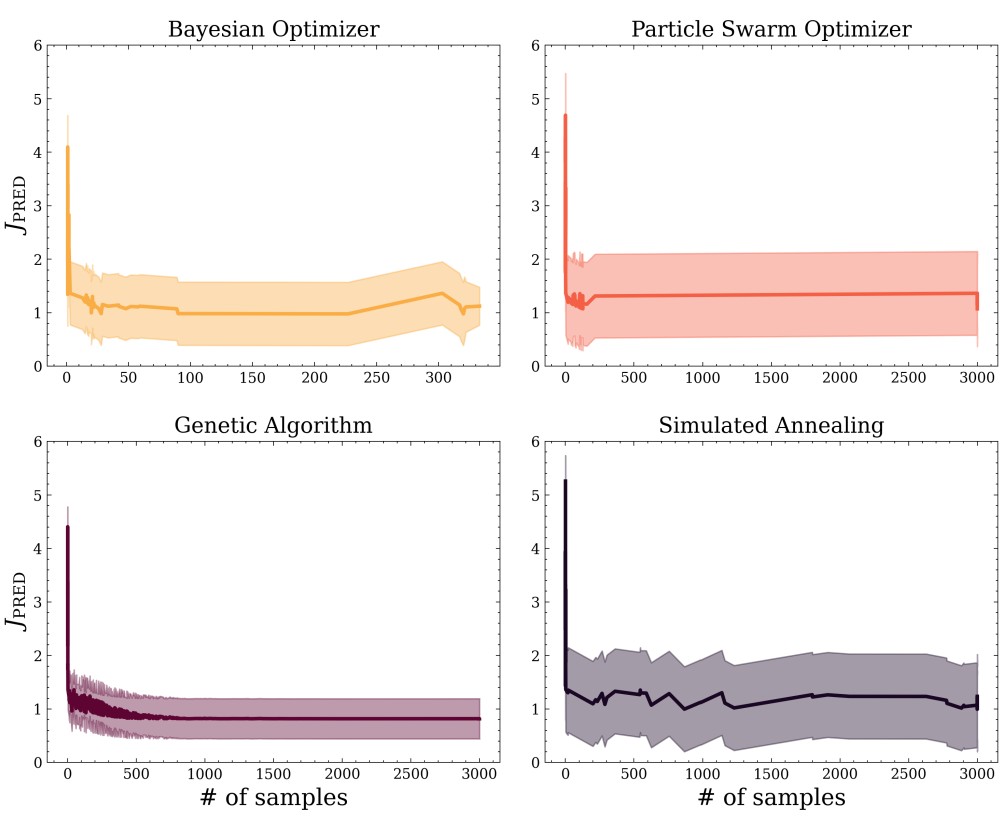

Figure 26: Evolution of the optimal solution with number of samples collected by solvers

Although the relative errors achieved by these solvers are comparable to our NODE approach, the data-driven optimization method has several significant limitations. First, data-driven optimization approach to determine the control profile requires discretization of the control variables over time,

which leads to a high-dimensional optimization problem. In many cases, a finer discretization is necessary to capture the system dynamics accurately, resulting in an increased number of decision variables and, consequently, a higher number of samples needed to locate the optimal solution. This escalates the computational burden, especially for high-fidelity simulations, as it necessitates running the simulator numerous times, potentially rendering the approach computationally expensive.

Second, data-driven solvers rely on the availability of the simulation model during the optimization process. However, this may not always be feasible; in some scenarios, only static or pre-collected data is accessible, which hinders the applicability of these solvers. Without the ability to query the simulator, the optimization cannot proceed effectively.

Finally, many practical optimization problems involve implicit and explicit constraints that must be satisfied. Most data-driven optimization algorithms lack robust mechanisms to incorporate such constraints into the optimization process. This limitation makes it challenging to apply these solvers to constrained optimization problems, as they may violate critical system requirements or fail to find feasible solutions. These challenges highlight the need for alternative approaches that can operate with limited data availability, and accommodate complex constraints within the optimization framework.

## E CODE SNIPPETS: USING PYOMO.DAE AND OMLT

Below, we show the use of Pyomo.DAE to solve Case Study 1.

```python
m = ConcreteModel()

m.t = ContinuousSet(bounds=(0, 1))

m.x1 = Var(m.t, bounds=(0, 1))
m.x2 = Var(m.t, bounds=(0, 1))
m.u = Var(m.t, initialize=0)

m.x1dot = DerivativeVar(m.x1)
m.x2dot = DerivativeVar(m.x2)

m.obj = Objective(expr=m.x2[1])

def _x1dot(M, i):
    if i == 0:
        return Constraint.Skip
    return M.x1dot[i] == M.u[i]

m.x1dotcon = Constraint(m.t, rule=_x1dot)

def _x2dot(M, i):
    if i == 0:
        return Constraint.Skip
    return M.x2dot[i] == M.x1[i] ** 2 + M.u[i] ** 2

m.x2dotcon = Constraint(m.t, rule=_x2dot)

def _init(M):
    yield M.x1[0] == 1
    yield M.x2[0] == 0
    yield ConstraintList.End

m.init_conditions = ConstraintList(rule=_init)
discretizer = TransformationFactory('dae.collocation')
```

```
1404   discretizer.apply_to(m, nfe=10, ncp=2, scheme='LAGRANGE-RADAU')
1405
1406   solver = SolverFactory('ipopt')
1407
1408   results = solver.solve(m, tee=True)
1409
```

Next, we show the use of OMLT to embed a neural network and replace the constraints for differential equations.

```
1412   def _con_out1(m,t):
1413     if t==0:
1414       return Constraint.Skip
1415     return OM.dx1dt[t] == OM.nn[t].m.outputs[0]   #dx1dt
1416
1417   def _con_out2(m,t):
1418     if t==0:
1419       return Constraint.Skip
1420     return OM.dx2dt[t] == OM.nn[t].m.outputs[1]   #dx2dt
1421
1422   def _con_in1(m,t):
1423     #if t==0:
1424       # return Constraint.Skip
1425     return OM.x1[t] == OM.nn[t].m.inputs[1]   #x1
1426
1427   def _con_in2(m,t):
1428     #if t==0:
1429       # return Constraint.Skip
1430     return OM.x2[t] == OM.nn[t].m.inputs[2]   #x2
1431
1432   def _con_in3(m,t):
1433     #if t==0:
1434       # return Constraint.Skip
1435     return OM.u[t] == OM.nn[t].m.inputs[0]   #u
1436   OM=ConcreteModel()
1437
1438   #OM.nn=OmltBlock()
1439
1440   #OM.tf=Param(initialize=1)
1441   OM.t =ContinuousSet(bounds=(0,1))
1442
1443   OM.u=Var(OM.t,initialize=0,bounds=(-3,1))
1444   OM.x1=Var(OM.t,bounds=(0, 1))
1445   OM.x2=Var(OM.t,bounds=(0, 1))
1446
1447   OM.dx1dt = DerivativeVar(OM.x1,wrt=OM.t)
1448   OM.dx2dt = DerivativeVar(OM.x2,wrt=OM.t)
1449
1450   OM.obj = Objective(expr=OM.x2[1])
1451
1452   discretizer = TransformationFactory('dae.collocation')
1453   discretizer.apply_to(OM, nfe=10, ncp=2, scheme='LAGRANGE-RADAU')
1454
1455   net=load_keras_sequential(model_tf)
1456   formulation = FullSpaceNNFormulation(net)
1457   #formulation=ReluComplementarityFormulation(net)
       OM.nn=Block(OM.t)
       for t in OM.t:
         OM.nn[t].m=OmltBlock()
         OM.nn[t].m.build_formulation(formulation)

       OM.nn[t].m.out1=Constraint(OM.t, rule=_con_out1)
```

```
OM.nn[t].out2=Constraint(OM.t, rule=_con_out2)
OM.nn[t].in1=Constraint(OM.t, rule=_con_in1)
OM.nn[t].in2=Constraint(OM.t, rule=_con_in2)
OM.nn[t].in3=Constraint(OM.t, rule=_con_in3)

solver = SolverFactory('ipopt')

results = solver.solve(OM, tee=True)
```

