# OpenReview forum: "Incorporating Neural ODEs into DAE-Constrained Optimization Problems"
_ICLR.cc/2025/Conference — ICLR 2025 Conference Withdrawn Submission_

### Official Review · Reviewer_ois6 · 2024-10-25

**Soundness:** 3
**Presentation:** 2
**Contribution:** 3
**Rating:** 5
**Confidence:** 4

**Summary:**

The paper "Incorporating Neural ODEs into DAE-Constrained Optimization Problems" presents a framework that integrates Neural Ordinary Differential Equations (NODEs) with differential-algebraic equation (DAE) constrained optimization. The authors address the challenge of optimizing dynamic systems when underlying equations are unknown or partially known, by using NODEs as data-driven surrogates for the system dynamics.

Through a series of case studies, ranging from biochemical reactors to epidemiological modeling, the paper demonstrates how NODE-ADOpt can effectively solve complex, non-linear optimization problems. The authors also compare their approach with existing methods like SINDy and black-box optimization techniques, highlighting scenarios where NODE-ADOpt outperforms these strategies.

**Strengths:**

- The framework removes limitations from prior optimization strategies by allowing the incorporation of data-driven components, which can be particularly useful when dealing with complex, non-linear, and noisy systems.
- Multiple experiments are conducted to validate the proposed method, and the results are systematically compared with existing approaches like SINDy and black-box optimization. This comparison highlights the advantages and performance improvements achieved by the proposed framework
- The framework's ability to outperform existing strategies under certain conditions makes it a promising candidate for real-world applications, offering both improved performance and flexibility.

**Weaknesses:**

- Broaden the literature review to include a more comprehensive discussion of alternative approaches, particularly those that integrate machine learning with optimization. Highlight where NODE-ADOpt stands in relation to these methods, its unique advantages, and any scenarios where alternative methods might be preferred.
- Although the paper mentions the tuning of hyperparameters, it lacks a discussion or strategy for optimizing them. This aspect is crucial, as the performance of neural networks can be highly sensitive to hyperparameter choices, especially in the context of dynamic optimization. It is also recommended to mention the hyperparameters that are used in each case for repudicibilty.
-It is recommended to include additional experiments that evaluate the performance of NODE-ADOpt on datasets with varying noise levels, different initial conditions, and unseen scenarios. Provide insights on regularization techniques, training strategies, or architectural adjustments that could improve generalization.
- While detailed, some mathematical explanations may be difficult to follow. It is highly recommended to explain elaborate your approach more clearly

**Questions:**

I added the recommendation in the box above

---

### Official Review · Reviewer_QyES · 2024-11-01

**Soundness:** 2
**Presentation:** 2
**Contribution:** 2
**Rating:** 3
**Confidence:** 4

**Summary:**

This paper introduces NODEs as surrogates for constrained dynamic optimization problems, and numerically tests their effectiveness on several study cases, showing improved performance over existing strategies such as SINDy.

**Strengths:**

The overall clarity and simplicity are good.

**Weaknesses:**

1. The significance of contribution.  The idea of using a neural network as a surrogate in control is not novel. What is the necessity and contribution of using a particular network architecture (i.e. NODE) in control problems?
1. The quality of baselines. The only compared baseline is SINDy, which is designed mainly for symbolic discovery but not for dynamics modeling or control. Does the proposed NODE model also have the ability to discover symbolic expression? If not, then SINDy might not be a suitable baseline. Maybe the commonly used network architectures such as MLP/ResNet/Transformer are the counterparts of NODE.

**Questions:**

See the 'Weaknesses' part.

---

### Official Review · Reviewer_j6Cu · 2024-11-02

**Soundness:** 2
**Presentation:** 2
**Contribution:** 1
**Rating:** 3
**Confidence:** 4

**Summary:**

The paper proposes a learning-based approach for solving differential algebraic equations (DAE) with data-driven components. It shows that pretrained neural ODE models can be integrated within an optimization problem with dynamic constraints which take the form of differential equations, by using collocation-based methods. In this setting, neural ODEs replace traditional numerical differential solvers, which can suffer of data quality, e.g. noise, partially unknown dynamic model and non linearities.

**Strengths:**

In settings where the dynamic model is partially unknown and/or when only data are available, neural ODEs model can effectively substitute a numerical differential solver, which would be incompatible.

**Weaknesses:**

The contribution lacks novelty in both methodology and technicality. Neural ODEs are well known to be able to capture the system dynamics from data [1]. While the authors recognize that in data-driven settings neural ODEs can be preferable over traditional numerical differential solvers, in my view it does not constitute enough contribution. The whole paper is based upon this sole idea, and as such it inevitably lacks of technicality. Section 2.2, which is the core section of the paper, as it formalizes how the system dynamic equations can be written using neural ODEs for use in a collocation-based method, provides only a simple example of how that can be done, lacking of generality.

[1] P.Kidger, On Neural Differential Equations, arxiv.org/abs/2202.02435

**Questions:**

Since I think that the authors' idea lacks of novelty for a contribution, in my view it is difficult to significantly improve this paper, but the current structure could be nonetheless improved: as anticipated above section 2.2 needs to provide a much more solid framework for using NODEs in a collocation based method in data-driven setting, which would strength the paper. The paper dedicates much attention to the impact of different hyperparameterization and training aspects of NODEs on how the dynamics are captured, but in my opinion it does not provide a valuable aspect, given that hyperparameters tuning is also a common practice in ML.

---

### Official Review · Reviewer_T2Th · 2024-11-03

**Soundness:** 2
**Presentation:** 2
**Contribution:** 2
**Rating:** 3
**Confidence:** 4

**Summary:**

This paper addresses optimization challenges in dynamic systems governed by differential-algebraic equations (DAEs), which are widely applied in fields such as process control, flight optimization, and epidemiology. Traditional optimization methods, like single shooting, multiple shooting, and direct transcription, work well with known mechanistic models but struggle when models are incomplete or don’t match empirical data. This paper integrates machine learning with DAE optimization by embedding Neural ODEs into the optimization process. Numerical examples include biochemical reactor control and epidemic modeling. The authors compare this method against  Sparse identification of non-linear dynamics (SINDy) method. The study also formalizes an optimization framework for DAEs with neural network components and introduces representations for standard neural network architectures.

**Strengths:**

This is an interesting paper about integrating algebraic constraints into neural ordinary differential equations. I believe the authors pose an important research question.

**Weaknesses:**

Formal aspects:
- there are typos in the paper or missing lines - see line 094
- authors should use both \cite and \citep as appropriate
- some figures do not include units, some lack axis labels

Prior work:
- the authors do not cite many similar publications from the recent literature. I encourage authors to search for keywords such as neural DAEs, or physics-informed neural networks for DAEs
Suggested related works include:
https://link.springer.com/article/10.1007/s00521-022-07886-y
https://ieeexplore.ieee.org/document/10669810
https://ieeexplore.ieee.org/document/9844253

Technical aspects:
- the limitations of this work are missing
- the collocation method presented in section 2.2 is not novel in itself
- the case studies are very simple, low-dimensional academic examples that can be easily solved with existing tools
I encourage the authors to benchmark the proposed method against state-of-the-art DAE solvers, for instance, those in Julia and Matlab
https://docs.sciml.ai/DiffEqDocs/stable/tutorials/dae_example/
https://www.mathworks.com/help/matlab/math/solve-differential-algebraic-equations-daes.html
- the paper fails to convince that it presents state of the art method dealing with problems that are currently beyond the reach of existing methods

**Questions:**

What type of DAE systems can tackled with the proposed method? The discussion on stiffness, and index of the DAE would be very valuable for the reader.
The comparison with SINDY is poorly motivated and seems a bit arbitrary; why not compare it with other nonlinear system identification techniques, such as the deep Koopman operator method or neural state space models?

---

### Note · Authors · 2024-11-27

I have read and agree with the venue's withdrawal policy on behalf of myself and my co-authors.